# PNPLA3 allele frequency has no impact on biliary bile acid composition or disease course in patients with primary sclerosing cholangitis

**Martti Färkkilä[1]\*, Hannu Kautiainen[2,3], Andrea Tenca[1], Kalle Jokelainen[1], Johanna Arola[4]**

**1** Department of Gastroenterology, University of Helsinki and Helsinki University Hospital, Helsinki, Finland,
**2** Folkhälsan Research Center, Helsinki, Finland, **3** Institute of Public Health and Clinical Nutrition, University
of Eastern Finland, Kuopio, Finland, **4** Department of Pathology, University of Helsinki and Helsinki University
Hospital, Helsinki, Finland

\* Martti.Farkkila@hus.fi

## Abstract

### Background and aims

Primary sclerosing cholangitis (PSC) is a chronic inflammatory disease that leads to bile
duct strictures, cholestasis, and biliary cirrhosis. PNPLA3 (patatin-like phospholipase
domain containing 3), regulates cellular lipid synthesis by converting lysophosphatidic
acid into phosphatidic acid. Isoleucine mutation to methionine at position 148 (I148M)
causes a loss of this function. Only two studies, with contradictory results, have evaluated
the role of PNPLA3 in PSC. The rs738409(G) variant of PNPLA3 has been associated
with an increased risk for transplantation in male patients with dominant strictures (DS).
The study aimed to evaluate the PNPLA3 allele frequency effect on the clinical outcomes,
progression, and prognosis of PSC. Furthermore, we analyzed the impact of PNPLA3 on
phospholipid and bile acid composition to evaluate the effect of the PNPLA3 status on
UDCA response.

### Patients and methods

We recruited 560 patients prospectively and collected clinical and laboratory data as well as
liver histology and imaging findings. PNPLA3 (CC, CG, GG) alleles were analyzed with Taq-
ManTM. We also analyzed bile acids (BA), cholesterol and phospholipids and individual BA
from a sample aspirated during endoscopic retrograde cholangiography (ERC).

### Results

Among the recruited patients, 58.4%, 35.7% and 5.9% had the wild (CC), heterozygous
(CG) and homozygous (GG) alleles, respectively. The PNPLA3 haplotype did not impact
bile composition or individual BA. In addition, we found no differences in age at diagnosis,
disease progression, liver fibrosis or survival between the cohorts.

pone.0277084

Bio-Medico di Roma, ITALY

**Data Availability Statement:** All relevant data are
within the paper.

**Funding:** Financial support: The study was supported by a grant from State funding for University-level health research TYH2016208 (MF). https://hussote.sharepoint.com/sites/00004/Tutkimusrahoitus/Sivut/default.aspx The funders had no role in study design, data collection and analysis, decision to publish, or preparation of the manuscript.

**Competing interests:** The authors have declared that no competing interests exist.

**Abbreviations:** AUC, area under curve; BA, bile acids; CA, cholic acid; CCA, cholangiocarcinoma; CDCA, chenodeoxycholic acid; DCA, deoxycholic acid; DS, dominant stricture; ERC, endoscopic retrograde cholangiography; ERCtwAUC, ERC time-weighted scores, area under curve; FXR, farnesoid X receptor; GLC, gas liquid chromatography; HCC, hepatocellular carcinoma; IBD, inflammatory bowel disease; LCA, lithocholic acid; mM, milliemolar; MRCP, magnetic resonance cholagio pancreatograph; P-ALP, plasma alkaline phosphatase; P-AST, plasma aspartate aminotransferase; P-GT, P-gammaglutamyl transferase; PNPLA3, patatin-like phospholipase domain containing 3; PSC, Primary sclerosing cholangitis; SMR, standardized mortality rate; UDCA, ursodeoxycholic acid; UNL, upper normal limit.

## Conclusions

The PNPLA3 I148M variant had no significant impact on on bile composition, including UDCA content, clinical outcomes, progression of liver fibrosis, hepatobiliary cancer risk, liver transplantation, or overall survival.

## Introduction

Primary sclerosing cholangitis (PSC) is a chronic inflammatory disease of the biliary epithelium leading to strictures and eventually biliary cirrhosis [1]. The clinical course of the disease is variable and unpredictable.

The etiopathogenesis of PSC is unknown. The disease is regarded a heterogeneous disorder with genetic, immunologic, and environmental factors [1, 2]. The risk of PSC is significantly increased in offsprings, siblings, and parents of the patients with the HR 11.5 (1.6–84.4), 11.1 (3.3–37.8), and 2.3 (0.9–6.1), respectively [3], suggesting a clear genetic predisposition. Several case–controlled genome-wide association studies have revealed ≥20 susceptibility loci [4], the strongest association being with the HLA complex [5]. The role of PNPLA3 (patatin-like phospholipase domain containing protein 3) in the pathogenesis and disease progression of several liver diseases, including PSC has also been studied [6]. PNPLA3 regulates cellular lipid synthesis by converting lysophosphatidic acid into phosphatidic acid. Isoleucine mutation to methionine at position 148 (I148M) of PNPLA3 causes a loss of this function leading to increased triglyceride synthesis and accumulation in the liver [7].

PNPLA3 mRNA is expressed in hepatic stellate cells [8]. PNPLA3 rs738409 C>G p.I148M is associated with steatosis and fibrosis and an increased risk of cirrhosis and hepatocellular cancer [6]. Additionally, the PNPLA3 variant has been associated with elevated of transaminases in general population [9] and in patients with IBD [10].

Little is known about the role of this PNPLA3 variant in regulating bile acid composition and phospholipid metabolism. Chen F et al. [11] analyzed fibroblast growth factor (FGF19) levels, a surrogate marker of farsenoid X receptor (FXR) activity in lean and obese patients with NAFLD. They found that FGF19 levels were significantly higher in lean patients, but the prevalence of PNPLA3 GG polymorphism was similar, suggesting that the rs738409 variant did not impact FXR activity. In addition, PNPLA3 has shown to promote phospholipid synthesis in mammalian cells [7]. The deficiencies of a biliary bicarbonate "umbrella", and the loss of alkalization of cholangiocyte apical membrane proximity have been suggested to increase membrane permeability of toxic BA leading to bile duct injury, inflammation, and strictures in PSC [12, 13]. The integrity of the cholangiocyte apical glycocalyx appears to be critical for cholangiocyte protection [14]. The apical bile salt receptor, TGR5, and the glycocalyx stabilizing enzyme fucosyltransferase 2 [1, 15, 16] protect cholangiocytes against BA. Alterations in the biliary phospholipid composition in the pathogenesis of PSC are based on the Mdr2 gene knockout mice studies, shown to develop cholangitis spontaneously [17].

A study of the role of the rs738409 variant in PSC including two patient cohorts [18], demonstrated that the common I148M variant of the PNPLA3 gene was a risk factor for decreased survival. However, this was observed only in male PSC patients with advanced disease and dominant stricture (DS) requiring endoscopic intervention. Moreover, the variant was not significantly associated with the development of DS [18]. Another study included 178 patients with PSC and evaluated the impact of PNPLA3 p.I148M on liver injury in cholestatic liver diseases. It was observed that an increasing number of risk alleles had no impact on transaminases, the risk of cirrhosis, or the need for liver transplantation [19].

Ursodeoxycholic acid (UDCA), has widely been used to treat PSC patients with reduction of cholestatic liver enzymes. However, only 30–67% of patients respond to UDCA assessed by the decrease or normalization of serum-alkaline phosphatase (S-ALP), [20–22]. Patients treated with UDCA, achieving a persistent improvement of S-ALP to <1.5 UNL, have significantly longer end-points- free survival [22]. However, UDCA has not been demonstrated to affect liver transplantation-free survival or liver-related death [23].

We evaluated the effect of PNPLA3 allele frequency on the clinical outcomes, disease progression and survival in a large population of PSC patients. Additionally, we analyzed the impact of PNPLA3 on biliary cholesterol, phospholipid, and BA composition in UDCA treatment-naïve patients and those on UDCA therapy to determine whether the PNPLA3 had an impact on UDCA response.

## Material and methods

### Patients

In total, 560 patients from the Helsinki University Hospital PSC registry based on hospital records were prospectively enrolled from 2010–2016 with follow-up till 2020. The diagnosis was based on the EASL Clinical Practice Guidelines [24]. All the patients underwent endoscopic retrograde cholangiography (ERC), and colonoscopy. Patients' demographic, clinical and laboratory data, as well as liver histology and imaging findings, were collected (Table 1). Furthermore, 82% and 69% of male, and female patients, respectively, were diagnosed with IBD. Liver function tests were performed either a day before or on the day of ERC. For analysis of the I148M PNPLA3 polymorphism (rs738409), DNA was isolated from whole-blood samples using standard procedures. Additionally, PNPLA3 (CC, CG, GG) alleles were analyzed with TaqManTM SNP Genotyping Assay (Applied Biosystems, Foster City, CA, USA) [25]. A composite end point consisting of CCA, hepatocellular carcinoma (HCC), liver transplantation and deaths was created as a clinical outcome.

### ERC examinations and bile sample collection

The indications for ERC were 1) documentation of diagnosis due to elevated P-ALP levels in conjunction with IBD, 2) magnetic resonance cholangiography (MRCP) or liver biopsy suggestive of PSC, and 3) surveillance of disease progression and biliary dysplasia [24]. The ERC based diagnosis and dysplasia surveillance strategy at Helsinki University Hospital has been previously presented [26, 27]. Images were evaluated using the modified Amsterdam score [26]. DS observed during ERC was defined as a stenosis with a diameter of ≤1.5 mm of the common duct or ≤1.0 mm of a hepatic duct within 2 cm of the bifurcation. A dilatation was made when DS was diagnosed, or the cytology brush could not be passed through the stenosis.

### Brush cytology

Brush cytology was collected from extra- and intrahepatic bile ducts, regardless of DS [26] using a brush with a guide wire (RX Cytology Brush, Boston Scientific, MA, USA).

### Biliary bile samples

A bile sample was aspirated using a balloon catheter, immersed into liquid nitrogen, and stored at -80 C˚. Biliary BA, phospholipid and cholesterol composition were analyzed in a subgroup of 220 patients, with 56 UDCA treatment naïve patients and 164 patients on UDCA therapy. Bile acids were silylated to trimethyl silylethers and quantitated by gas-liquid chromatography (GLC) on the SE-30 capillary column and the individual BA content was analyzed

**Table 1. Clinical characteristics of patients based on PNPLA3 haplotype.**

| Variables | PNPLA3 rs738409 | | | p for linearity |
|---|---|---|---|---|
| | CC N = 327 (58.4%) | CG N = 200 (35.7%) | GG N = 33 (5.9%) | |
| Number of males, (%) | 192(59) | 126(63) | 18(55) | 0.73 |
| Age at PSC diagnosis, mean (SD) | 37(14) | 36(13) | 36(13) | 0.20 |
| IBD, n (%) | 240(73) | 162(81) | 24(73) | 0.21 |
| BMI, kg/m$^2$,mean (SD) | 25.6(4.2) | 25.6(4.7) | 26.2(4.4) | 0.53 |
| Age at diagnosis IBD, mean (SD) | 27(12) | 26(11) | 29(12) | 0.98 |
| ERC score, median (IQR) | 5 (2,9) | 4 (2,8) | 4 (2,7) | 0.20 |
| Histology, stage, number | 146 | 85 | 16 | 0.67 |
| 0 | 62(42) | 40(47) | 7(44) | |
| 1 | 33(23) | 20(24) | 4(25) | |
| 2 | 36(25) | 13(15) | 3(19) | |
| 3 | 10(7) | 8(9) | 1(6) | |
| 4 | 5(3) | 4(5) | 1(6) | |
| Histology, grade, number, (%) | 150 (58) | 90 (35) | 18 (7) | 0.44 |
| 0 | 89(59) | 63(70) | 10(56) | |
| 1 | 42(28) | 18(20) | 5(28) | |
| 2 | 10(7) | 6(7) | 3(17) | |
| 3–4 | 9(6) | 3(3) | 0(0) | |
| P-ALP, U/l, median (IQR) | 138 (95,234) | 132 (98,219) | 146 (115,209) | 0.92 |
| P-GGT, U/l, median (IQR) | 90 (34,248) | 131 (43,265) | 178 (66,272) | 0.031 |
| P-ALT, U/l, median (IQR) | 43 (25,79) | 48 (24,97) | 51 (33,85) | 0.19 |
| P-AST, U/l, median (IQR) | 35 (26,57) | 39 (27,59) | 50 (33,83) | 0.016 |
| P-ALB, g/l | 38 (35,41) | 38 (36,41) | 39 (36,41) | 0.23 |
| P-Bilirubin, μmol/l, median (IQR) | 11 (7,16) | 11 (8,17) | 11 (7,16) | 0.69 |
| B-Platelets, E9/l, median (IQR) | 254 (205,314) | 252 (210,307) | 251 (212,312) | 0.99 |
| APRI, median (IQR) | 0.36 (0.23,0.63) | 0.39 (0.26,0.62) | 0.46 (0.28,0.77) | 0.074 |
| FIB4, median (IQR) | 1.15 (0.77,1.75) | 1.10 (0.75,1.77) | 1.10 (0.95,1.82) | 0.78 |

Abbreviations: PSC, primary sclerosing cholangitis; IBD, inflammatory bowel disease; BMI, body mass index; ERC, endoscopic retrograde cholangiography; P-ALP, alkaline phosphatase; P-GGT, gamma glutamyl transferase; P-ALT, alanine aminotransferase; P-AST, aspartate aminotransferase; P-ALB, albumin; APRI, AST-platelet ratio index.

[28]. The amount of BA was expressed as content (mM) and molar percentages (mM%); 5-α cholestane was used as an internal standard and phospholipids were analyzed enzymatically with commercial kits (Enzabil-kit; Nycomed).

## Statistics

The descriptive statistics were presented as means with standard deviation (SD), as medians with interquartile range (IQR) or as counts with percentages. Group differences were evaluated using unpaired Student's t-test, Mann-Whitney U test, chi-squared test. Statistical significances for the hypothesis of linearity of clinical characteristics across PNPLA3 haplotype levels of clinical characteristics was evaluated using the Cochran-Armitage (chi-squared) test for trend, the Cuzick test, ordered logistic regression models and analysis of variance with appropriate contrast. Kaplan-Meier's survival analysis was performed to estimate the cumulative survivals (Fig 1) and survival for composite end point (Fig 2): CCA, HCC, liver transplantation or liver related deaths; adjusted cumulative rates were estimated using two propensity score-based techniques, stratification, and weighting (MMWS, marginal mean weighting through

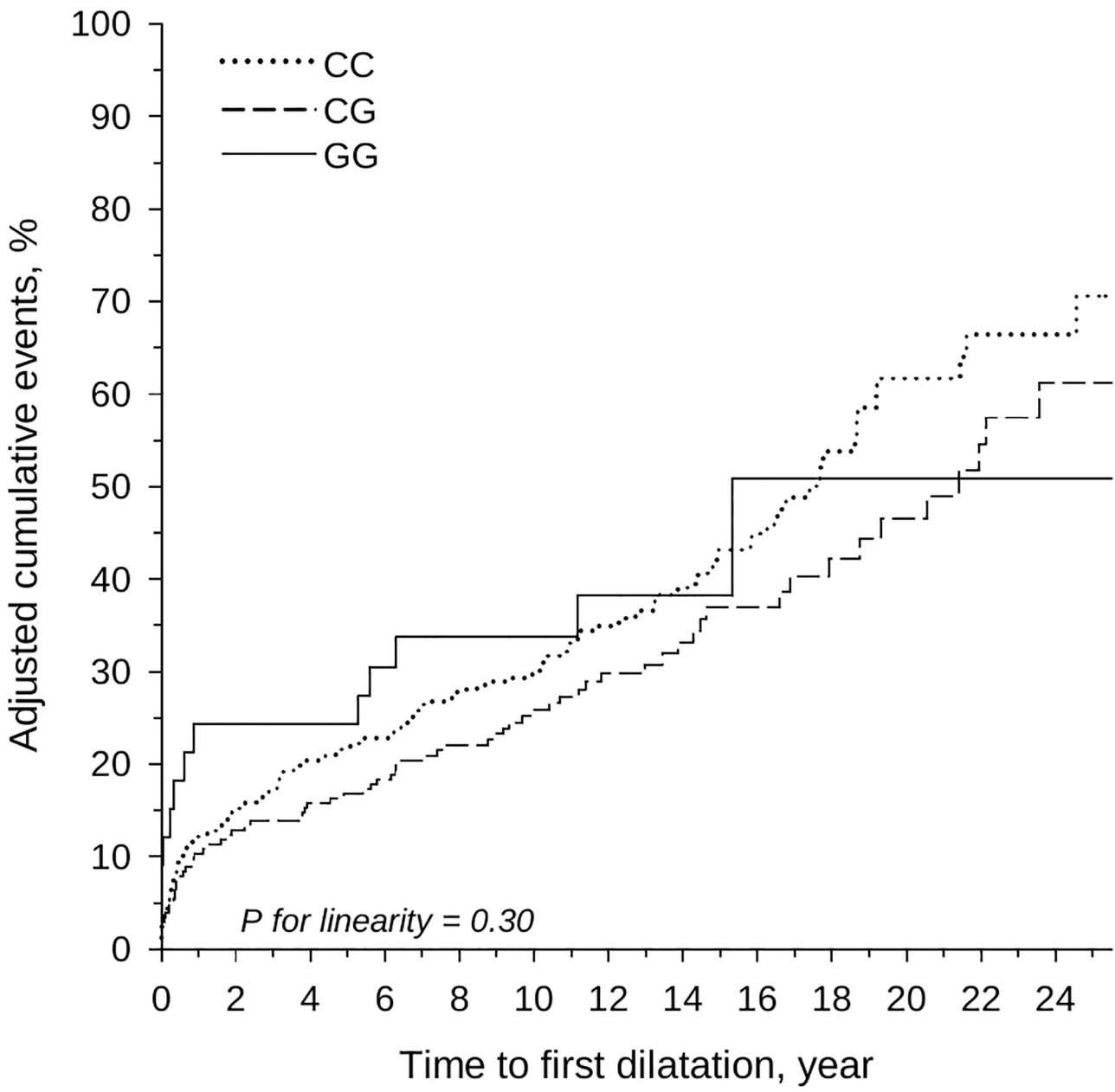

**Fig 1. Impact of PNPLA3 haplotype on the development of strictures assessed by the time to first dilatation.** Impact of the PNPLA3 haplotype; adjusted for sex and age at PSC diagnosis.

stratification) [29]. MMWS is an extension of propensity score matching that combines propensity score stratification and inverse probability of treatment weighting. Overall cumulative rate relationships were investigated using the log-rank test or Cox proportional hazards models. The impact of PNPLA3 allele frequency and UDCA administration and their interaction were evaluated using two-way rank-based analysis of covariance (ANCOVA) factorial models [30]. The standardized mortality rate (SMR), which is the ratio between observed and expected numbers, was calculated using subject-year methods with 95% CIs, assuming a Poisson distribution.

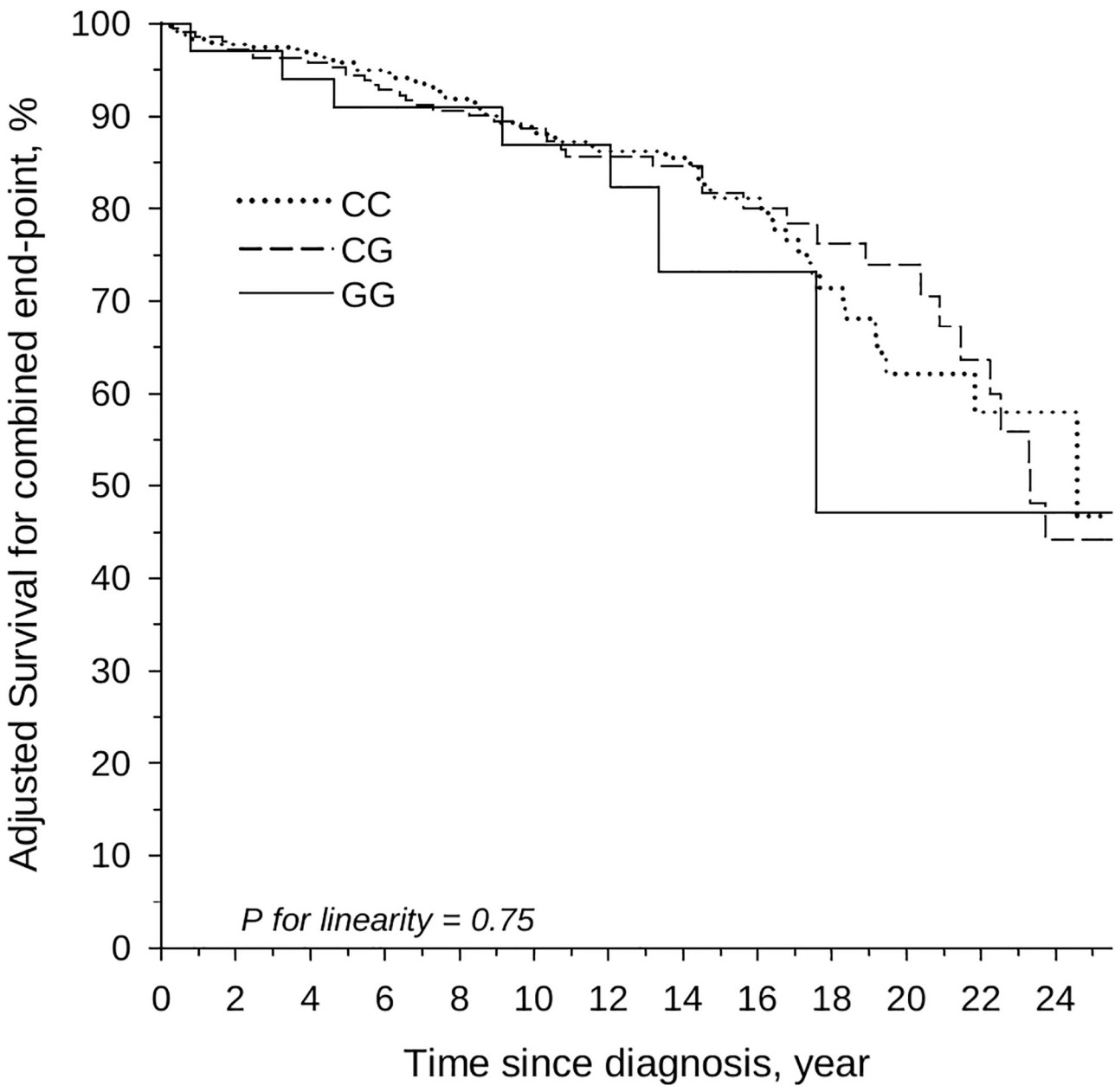

**Fig 2. Proportion of patients who did not reach the composite end point (CCA, liver transplantation, and deaths) according to the PNPLA3 allele frequency; adjusted for sex and age at PSC diagnosis.**

Furthermore, we calculated areas under the curve (AUC) with the trapezoidal method in terms of longitudinal ERC-score (ERCtwAUC). In case of violation of the assumptions, a bootstrap-type method was used (10 000 replications) to estimate 95% CI. All analyses were conducted using Stata (version 17.0, Stata Corp, College Station, TX, USA).

## Ethics

All the patients included in the PSC registry provided a written informed consent. The study was performed following the principles of GCP and in accordance with the ethical guidelines

of the Declaration of Helsinki (6th revision, 2008). The study protocol was approved by Helsinki University Hospital Ethical Committee IV, HUS/1566/2020.

## Results

### PNPLA3 allele frequency on the clinical course of the disease

Of the 560 patients, 327 (58.4%), 200 (35.7%), and 33 (5.9%) patients had the wild type (CC), heterozygous (CG) and 33 homozygous (GG) alleles, respectively. The follow-up time was 3990 person-years (mean 12.2 years), 2528 years (12.6 years) and 383 years (11.6 years), for patients with the CC, CG and GG haplotype, respectively. Clinical and laboratory data are presented in Table 1. No relationship in age at PSC diagnosis, median ERC score, or liver histology was observed between the cohorts grouped by the increasing allele frequency. In addition, plasma aspartate aminotransferase (P-AST) and P-gamma-glutamyl transferase (P-GT) were significantly increased with the increasing allele frequency. The PNPLA3 haplotype was not associated with liver fibrosis assessed by either APRI- and Fib4-score or the proportion of patients with advanced fibrosis (Metavir 3–4), (Table 1).

### PNPLA3 haplotype and progression of bile duct changes at ERC

The progression of strictures needing dilatation was assessed by the time to first dilatation. In total, at least one dilatation was performed in 223 patients, 137, 73, and 13 patients with CC, CG and GG, respectively, during the follow-up period No relationship was observed between PNPLA3 haplotype and the development of strictures needing dilatation (p for linearity 0.30). The crude prevalence of patients needing dilatation at least once was 71% (95% CI: 59 to 82), 62% (49 to 75), and 50% (28 to 77) in patients with CC, CG, and GG alleles, respectively, for the 25-year follow-up, (p for linearity 0.29). Fig 1 shows the age at diagnosis and sex-adjusted time to first dilatation. The total ERC-scores and time weight AUC of ERC did not demonstrate any relationship with the increasing PNPLA3 allele frequency: (median 6.0, IQR 2.3,9.0), 5.4 (2.0,8.5), CG, and 5.4 (2.5,8.7) in CC, CG, and GG, respectively.

### Development of hepatobiliary cancers and mortality

CCA or HCC (Table 2) was diagnosed in 35 patients during the follow-up; 23 (7%), 12(6%) and in no patients with CC, CG and GG haplotype, respectively, (p for linearity 0.19; adjusted for sex, age and IBD). In total, 76 patients underwent liver transplantation during the follow-up period: 40 (12.2%), 29 (14.5%), and 7 (21.2%) in CC, CG and GG-allele, respectively (p for linearity = 0.35 adjusted for sex, age and IBD). Fig 2 shows the overall survival, which is the proportion of patients that did not reach the composite end point; no relationship was observed between the groups (p for linearity 0.75). The 25-years crude survival rate was 48% (95% CI: 23–66), 48% (95% CI: 31 to 64), and 49 (95% CI: 0 to 83) in CC, CG, and GG haplotype, respectively (p for linearity = 0.88).

In the study population, 32 patients died, and the most common cause of death was CCA. No differences in mortality were observed between the groups (Table 2). The SMR in patients with CC, CG and GG haplotype was 1.40 (95%CI 0.92–2.15), 1,12 (95%CI 0,58–2,15), and 1.95 (95%CI 0,49–7,79), respectively.

### Effect of UDCA therapy and PNPLA3 haplotype on bile composition

We evaluated the impact of PNPLA3 haplotype and UDCA administration on total biliary BA, cholesterol, and phospholipid composition in UDCA treatment naïve patients and those receiving UDCA therapy, (Table 3). The mean UDCA dose/kg (SD) was 17.1 (4.4), and the

**Table 2. Effect of PNPLA3 allele frequency on study population's crude 25-year survival rate and significant outcomes.**

| Variables | | PNPLA3 rs738409 | | | p for linearity** |
|---|---|---|---|---|---|
| | Total (n) | CC | CG | GG | |
| | | N = 327 | N = 200 | N = 33 | |
| CCA | 25 | 16 | 9 | 0 | 0.20 |
| Survival,% (95% CI)* | | 92 (87 to 96) | 95 (90 to 97) | 100 (..) | |
| HCC | 11 | 8 | 3 | 0 | 0.21 |
| Survival,% (95% CI)* | | 91 (78 to 97) | 94 (76 to 98) | 100 (..) | |
| Liver transplantation, n (%) | 76 | 40 | 29 | 7 | 0.35 |
| Survival, (95% CI)* | | 56 (27 to 75) | 51 (35 to 69) | 49 (0 to 85) | |
| Deaths, n (%) | 32 | 21 | 9 | 2 | 0.55 |
| Survival,% (95% CI)* | | 88 (81 to 93) | 94 (89 to 98) | 94 (81 to 100) | |

Abbreviations: CCA, cholangiocarcinoma, HCC, hepatocellular carcinoma

*Bias corrected bootstrap estimation (5000 replications) were used to derive 95% confidence intervals.

** Adjusted for sex, age and IBD

median (IQR) 18 (15–20). In UDCA treatment-naïve patients, the PNPLA3 haplotype did not impact the bile composition. In contrast, primary BA and deoxycholic acid (DCA) levels were decreased in patients receiving UDCA therapy; however, UDCA administration did not impact lithocholic acid levels. No interaction was observed between PNPLA3 haplotype and UDCA therapy on bile composition, Table 3).

**Table 3. Impact of PNPLA3 allele frequency and UDCA administration and their interaction on bile composition in PSC.**

| Variable | PNPLA3* | | | | | | Effect** | | |
|---|---|---|---|---|---|---|---|---|---|
| Haplotype | CC | | CG | | GG | | Main | | Inter-action |
| | UDCA | | UDCA | | UDCA | | Haplo | UDCA | |
| Use of UDCA | Naïve N = 27 Median (IQR) | Users N = 95 Median (IQR) | Naïve N = 23 Median (IQR) | Users N = 57 Median (IQR) | Naïve N = 6 Median (IQR) | Users N = 12 Median (IQR) | p-value | p-value | p-value |
| **Bile composition** | | | | | | | | | |
| Cholesterol, mM | 2.62 (1.89,3.30) | 1.57 (1.03,2.01) | 2.92 (2.26,4.70) | 1.28 (0.88,1.82) | 2.74 (1.45,3.90) | 1.17 (1.00,1.83) | 0.80 | <0.001 | 0.20 |
| Bile acids, mM | 20.4 (16.2,34.0) | 22.9 (16.5,28.3) | 23.5 (16.4,29.4) | 21.4 (14.7,26.6) | 25.8 (8.3,35.0) | 21.7 (16.7,27.1) | 0.97 | 0.36 | 0.72 |
| Phospholipids, mM | 8.3 (5.8,11.7) | 8.9 (6.1,11.3) | 9.4 (6.3,11.1) | 7.9 (5.3,10.5) | 11.1 (3.4,13.4) | 7.4 (6.3,10.9) | 0.78 | 0.34 | 0.87 |
| Cholesterol, mM% | 7.3 (6.0,10.3) | 4.4 (3.5,5.6) | 8.5 (6.4,12.0) | 4.0 (3.0,5.3) | 7.7 (7.2,10.7) | 4.6 (3.4,6.4) | 0.78 | <0.001 | 0.36 |
| Bile acids, mM% | 68 (62,70) | 69 (64,72) | 65 (62,69) | 69 (65,75) | 65 (61,67) | 68 (64,73) | 0.41 | 0.006 | 0.68 |
| Phospholipids, mM% | 25 (22,28) | 27 (23,30) | 25 (22,27) | 26 (22,29) | 26 (25,31) | 27 (23,30) | 0.18 | 0.49 | 0.63 |
| **Bile acids, mg** | | | | | | | | | |
| Lithocholic | 1.03 (0.65,2.32) | 1.56 (0.41,3.33) | 0.96 (0.45,1.45) | 0.96 (0.22,3.28) | 1.02 (0.54,1.15) | 0.94 (0.27,3.33) | 0.42 | 0.75 | 0.95 |
| Deoxycholic | 12.9 (3.2,24.1) | 5.0 (0.2,9.2) | 7.0 (2.0,18.6) | 3.6 (0.1,9.5) | 4.5 (1.1,10.2) | 2.3 (0.2,12.6) | 0.54 | 0.005 | 0.58 |
| Chenodeoxy cholic | 31.4 (24.1,36.1) | 14.8 (11.8,20.3) | 34.0 (24.9,36.9) | 15.5 (13.1,21.3) | 32.8 (13.9,37.5) | 15.8 (13.3,25.8) | 0.58 | <0.001 | 0.44 |
| Cholic | 48.7 (34.8,65.0) | 16.4 (11.9,23.1) | 55.0 (45.1,62.4) | 17.5 (13.0,25.4) | 56.2 (48.9,59.9) | 19.5 (12.6,32.9) | 0.48 | <0.001 | 0.33 |
| UDCA | 0.81 (0.56,1.95) | 58.4 (50.2,64.1) | 0.98 (0.46,2.56) | 57.1 (46.0,65.3) | 0.59 (0.51,1.38) | 50.5 (39.5,59.4) | 0.29 | <0.001 | 0.87 |

*Values expressed as median (IQR)

**Adjusted for age and sex.

Abbreviations: mM, millimolar; UDCA, ursodeoxycholic acid

## Discussion

We found that the PNPLA3 I148M variant did not significantly impact clinical outcomes, liver fibrosis, DS development, hepatobiliary cancer risk, need for transplantation, or survival. Neither did the increasing allele frequency had any impact on BA composition. UDCA therapy markedly reduced the primary BA; however, this was not reflected by changes in liver enzymes, such as P-ALP or bilirubin or disease progression.

Fat metabolism in liver may have a major impact on the pathophysiology of liver diseases. The PNPLA3 protein has lipase activity towards triglycerides in hepatocytes and retinyl esters in hepatic stellate cells [6]. The I148M substitution leads to a loss of this function, thus promoting triglyceride accumulation in hepatocytes. The PNPLA3 gene variant rs738409 C>G located on chromosome 22 has been associated with susceptibility to NAFLD [31, 32], development of fibrosis, increased risk of alcohol-related cirrhosis [6] and mortality from alcoholic hepatitis [33]. Moreover, rs738409(G) is a risk factor for hepatocellular cancer in alcoholic cirrhotic patients with a two-fold HCC risk [34]. PNPLA3 I148M variant exhibits increased LPAAT activity and leads to increased cellular lipid accumulation. It also promotes progression to chronic liver disease under a large variety of harmful stimuli for the liver such as cholestasis or bacterial endotoxemia as suggested in PSC. Primary hepatic stellate cells from patients with the PNPLA3-rs738409 GG variant displayed significantly higher expression and release of pro-inflammatory cytokine [35]. In addition, in patients with autoimmune hepatitis, the PNPLA3 I148M variant has been associated with the progression of liver disease despite steatosis being similar across all PNPLA3-rs738409 genotypes [36]. The function of PNPLA3 has been extensively studied, but the molecular mechanisms leading to fibrosis and carcinogenesis remain unclear [6]. I148M polymorphism has been suggested to represent a general modifier of fibrogenesis and a key player in liver disease progression [37]. Presently, there are no directly available data on the rs738409(G) variant and the expression of FXR, which is the most important regulator of bile acid metabolism, or CYP7A1 (cholesterol 7α hydroxylase) activity, which is the key enzyme of BA synthesis.

A recent study evaluating plasma BA profiles in PSC to predict hepatic decompensation, demonstrated that the risk was associated with increased concentration of total BA and conjugated fraction of many BA. In contrast, higher glycine: taurine conjugation ratios were protective [38]. Intestinal dysbiosis has been described in PSC patients with and without IBD [39]. The effect of the PNPLA3 rs738409 variant on gut microbiota has been evaluated only in a few studies. Monga Kravetz A et al. [40] demonstrated that this PNPLA3 variant caused an increased on Firmicutes/Bacteroides-ratio in obese youth with NAFLD.

Thus far, only two studies with contradictory results have evaluated the role of PNPLA3 in PSC [18, 19]. Furthermore, no studies have evaluated the impact of PNPLA3 allele frequency on bile composition or progression of bile duct changes or fibrosis. Friedrich et al. [18] demonstrated that male carriers of the I148M variant showed a significantly reduced survival free of liver transplantation, but only in the presence of a DS. They concluded that male carriers of the I148M variant could be regarded as a high-risk subgroup, regarding surveillance strategies and liver transplant allocation. Hence, testing for the PNPLA3 variant could improve therapeutic approaches in PSC. However, the conclusions were based on a cohort with only 23 males carrying the I148M genotype (CG or GG) with DS. In addition, the authors included a validation cohort of Norwegian PSC patients with endoscopic intervention, consisting of 32 males with the I148M variant. They demonstrated that the variant impaired actuarial survival free of liver transplantation in that cohort.

In our study including 70 patients with DS who carried the I148M variant, we could not confirm the results of Friedrich et al. [18]. The I148M variant did not impact bile duct disease

assessed by sequential ERC examinations, development of cirrhosis or need for transplantation. In line with our results, Kruk et al. [19] could not find any significant differences in the genotype distribution of PNPLA3 observed between patients with cirrhosis and without cirrhosis. Neither were serum liver enzyme activities modified by the presence of risk variants. The prevalence of PNPLA3 haplotype CG (MM) was 34.8% and GG 4% (n = 7) in their study, compared to the present study, with CG 40% (n = 200) and GG 5.9% (n = 33), respectively. It was reported that CG haplotype prevalence in the general Finnish population was around 36% and that of GG was around 6% [41]. Hence, in Finland the PNPLA3 (RS738409 C>G P. I148M) allele frequency in PSC did not differ from that of the general population, suggesting that it is unlikely that PNPLA3 haplotype would predispose individuals to PSC development.

## Strengths and limitations

A strength of the present study was the large patient population with sequential ERC examinations for evaluating bile duct disease progression and assessing liver fibrosis based on histology and noninvasive fibrosis markers. However, the study's limitation was the relatively low number of late end points such as liver transplantation, CCA, or deaths.

## Conclusions

The present study included 560 PSC patients with different disease states. We observed that the PNPLA3 I148M variant did not have any significant impact on clinical outcomes, progression of liver fibrosis, dominant stricture development, hepatobiliary cancer risk, the need for liver transplantation or patient survival. In addition, the allele frequency of PNPLA3 did not have any significant impact on bile composition. In contrast to the previous study [18] we could not identify rs738409 C>G p.I148M variant as a risk factor for PSC, and PNPLA3 genotyping does not seem to have any role in stratifying patients with PSC for surveillance.

## Acknowledgments

We acknowledge laboratory assistant, Leena Kaipiainen for bile analysis at Biomedicum Helsinki, POB 700, FI-00029 HUS, Helsinki, Finland, and study nurse Virpi Pelkonen for recruiting patients into the study.

## Author Contributions

**Conceptualization:** Martti Färkkilä, Hannu Kautiainen.

**Data curation:** Martti Färkkilä, Hannu Kautiainen, Andrea Tenca, Kalle Jokelainen.

**Formal analysis:** Martti Färkkilä, Hannu Kautiainen.

**Funding acquisition:** Martti Färkkilä.

**Investigation:** Martti Färkkilä, Andrea Tenca, Kalle Jokelainen, Johanna Arola.

**Methodology:** Martti Färkkilä, Hannu Kautiainen.

**Resources:** Martti Färkkilä.

**Software:** Hannu Kautiainen.

**Writing – original draft:** Martti Färkkilä, Hannu Kautiainen.

**Writing – review & editing:** Andrea Tenca, Kalle Jokelainen, Johanna Arola.

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
