## [Decision Letter · Decision Letter 0]

1 Jun 2022

PONE-D-21-38918The impact of PNPLA3 allele dose on biliary bile acid composition and disease course in patients with primary sclerosing cholangitisPLOS ONE

Dear Dr. Färkkillä,

Thank you for submitting your manuscript to PLOS ONE. After careful consideration, we feel that it has merit but does not fully meet PLOS ONE’s publication criteria as it currently stands. Therefore, we invite you to submit a revised version of the manuscript that addresses the points raised during the review process.

We look forward to receiving your revised manuscript.

Kind regards,

Antonio De Vincentis

Academic Editor

PLOS ONE

Journal Requirements:

Reviewers' comments:

Reviewer's Responses to Questions

**Comments to the Author**

1. Is the manuscript technically sound, and do the data support the conclusions?

Reviewer #1: Yes

Reviewer #2: Partly

2. Has the statistical analysis been performed appropriately and rigorously? 

Reviewer #1: Yes

Reviewer #2: No

3. Have the authors made all data underlying the findings in their manuscript fully available?

Reviewer #1: Yes

Reviewer #2: Yes

4. Is the manuscript presented in an intelligible fashion and written in standard English?

Reviewer #1: Yes

Reviewer #2: No

5. Review Comments to the Author

Reviewer #1: The title reflects inappropriately the content of the study, in which no impact of PNPLA3 allele dose was find neither on BA composition nor disease course in patients with PSC. Please rephrase.

The allele groups of patients are not balanced in numerosity: only 5.9% of patients were mutated.

Was there any difference between the wildtype group and the CG-GG group?

To the knowledge of this reviewer, the pathogenesis of PSC is still unknown: published evidences address the PSC development to defects in mechanisms protecting against bile acid toxicity. In addition, the strong relationship of PSC in patients with IBD support the hypothesis of pro-inflammatory microbial components to the portal circulation and the possibility of an antigenic trigger of microbial origin. In the discussion authors say that to date, no data is available on the PNPLA3 variants and the expression of FXR. Though, to the knowledge of this reviewer this field has been explored in NAFLD patients, also with the analysis of gut microbiome. (Chen F et al., Lean NAFLD: A Distinct Entity Shaped by Differential Metabolic Adaptation. Hepatology. 2020) It could be interesting to know whether a correlation between PNPLA3, FXR expression, BA composition and gut microbiome could be found in PSC patients.

The statement in the discussion should be amended.

Authors reported 73%, 81%, 73% of patients with associated IBD according to CC, CG, GG PNPLA3 haplotype. There is no description of the severity of the disease, nor the medical-surgical treatment. Could you present these data?

Authors presented in the results section that P-AST and P-GT were significantly increased with increasing allele dose. Could you give an interpretation of this finding in the discussion?

Reviewer #2: Dear authors, thanks for giving me the opportunity to review your manuscript.

In the manuscript, the authors aim at demonstrating how the presence of different alleles of PNPLA3 might influence clinical outcome and bile composition in patients with primary sclerosing cholangitis.

Despite the conflicting results of the other studies available in the literature and the numerous sample studied (not always available in case of rare diseases), in the reviewer's opinion, the rationale of the study is not convincing and, unsurprisingly the results confirm it. PNPLA3 has been implied in fibrogenesis and carcinogenesis and has been linked to hepatocarcinoma which is not a frequent tumor in PSC patients. Furthermore, PSC patients usually develop liver fibrosis as a consequence of cholestasis and in many cases, the fibrosis is confined to the bile ducts. The mechanisms by which PNPLA3 induces liver damage are not completely understood and, in the reviewer's opinion, the rationale for studying their alleles in PSC is weak. However, the evaluation of bile acid composition may be interesting in these patients but a lot of confounding factors might be involved in this analysis (i.e. microbiota composition and IBD activity) which are not taken into account.

Furthermore, there are many punctuation errors in the manuscript and imprecisions.

Introduction:

The introduction is not fluent and the rationale is not well explained.

I suggest rephrasing sentences in lines 76 and 81.

In line 88 I would add that the UDCA effect on LT-free survival and liver-related death has never been demonstrated.

Methods:

The part on statistical analysis is too long and full of detail with respect to the data shown in the results section.

You mention a "compound" outcome in the results but I suppose composite outcomes have never been mentioned in the methods section. I suggest specifying the endpoints in this section.

Results:

line 162, specify the unit of measure of median values in brackets.

line 175, time to the first dilation since when? diagnosis? ERC?

in table 2 I would have distinguished HCC from CCA, which is, in the reviewer's opinion, quite important to the aim of the manuscript.

An English editing is suggested.

6. PLOS authors have the option to publish the peer review history of their article (what does this mean?). If published, this will include your full peer review and any attached files.

Reviewer #1: No

Reviewer #2: No

---

## [Author Response · Author response to Decision Letter 0]

1 Sep 2022

Response to reviewers’ comments

PONE-D-21-38918

The impact of PNPLA3 allele dose on biliary bile acid composition and disease course in patients with primary sclerosing cholangitis

PLOS ONE

Reviewer #1: The title reflects inappropriately the content of the study, in which no impact of PNPLA3 allele dose was find neither on BA composition nor disease course in patients with PSC. Please rephrase.

Response: The title has been rephrased: PNPLA3 ALLELE FREQUENCY HAS NO IMPACT ON BILIARY BILE ACID COMPOSITION OR DISEASE COURSE IN PATIENTS WITH PRIMARY SCLEROSING CHOLANGITIS

The allele groups of patients are not balanced in numerosity: only 5.9% of patients were mutated.

Response: The prevalence of the GG haplotype in the general Finnish population is around 6%, which is the same as in PSC patients cohort. PSC is a relatively rare disease and balancing the study population numerosity for PNPLA3 (RS738409 C>G P.I148M) allele homozygosity to the wild type would require a PSC population of 5540 cases. In the study of Friedrich K et al (ref 17) the total number of GG homozygotes from Germany and Norway was (4+19) =23 and in that of Kruk B et al (18) was 7. So, the number homozygotes (n=33) in the present study exceeds that of the two earlier published cohorts.

Was there any difference between the wildtype group and the CG-GG group?

Response: For clinical and laboratory parameters Only P-GT was higher in GC+GG-group compared to CC, see kindly Supplementary table 1. Clinical characteristics of patients based on PNPLA3 haplotype.

For clinical outcomes: CCA, HCC, liver transplantation and liver related deaths no statistically significant differences were seen in CC compared to GC+GG, see kindly Supplementary table 2. PNPLA3 allele dose and significant outcomes of study population. Crude 25 years survival.

To the knowledge of this reviewer, the pathogenesis of PSC is still unknown: published evidences address the PSC development to defects in mechanisms protecting against bile acid toxicity. 

Response: We agree the referee, and this is referred in the introduction: ‘The BA toxicity and the deficiencies of a biliary bicarbonate ‘‘umbrella’’, the loss of alkalization of cholangiocyte apical membrane proximity has suggested to increase membrane permeability of toxic BA leading to bile duct injury, inflammation, and strictures in PSC (19,20). The integrity of the cholangiocyte apical glycocalyx appears to be critical for cholangiocyte protection (21).

In addition, the strong relationship of PSC in patients with IBD support the hypothesis of pro-inflammatory microbial components to the portal circulation and the possibility of an antigenic trigger of microbial origin. In the discussion authors say that to date, no data is available on the PNPLA3 variants and the expression of FXR. Though, to the knowledge of this reviewer this field has been explored in NAFLD patients, also with the analysis of gut microbiome. (Chen F et al., Lean NAFLD: A Distinct Entity Shaped by Differential Metabolic Adaptation. Hepatology. 2020) It could be interesting to know whether a correlation between PNPLA3, FXR expression, BA composition and gut microbiome could be found in PSC patients.

The statement in the discussion should be amended.

Response: Thank you for bringing up this important aspect. We have added a sentence regarding the FXR activity: Chen F et al. analyzed FGF19 levels, a surrogate marker of FXR activity in lean obese patients with NAFLD. ‘They found that FGF19 levels were significantly higher in lean patients, but the prevalence of PNPLA3 GG polymorphism was similar, suggesting that rs738409 variant does not have impact on FXR activity’. In addition, a comment regarding the role of PNPLA3 in gut microbiota was added.

Authors reported 73%, 81%, 73% of patients with associated IBD according to CC, CG, GG PNPLA3 haplotype. There is no description of the severity of the disease, nor the medical-surgical treatment. Could you present these data?

Response: The primary aim of the study was not to evaluate the role of PNPLA3 allele dose on treatment response or need for surgery in IBD. We have now made a secondary analysis of the impact of PNPLA3 haplotype on the activity and disease outcome of IBD based on PNPLA3 allele dose, see kindly Supplementary table 3.

Authors presented in the results section that P-AST and P-GT were significantly increased with increasing allele dose. Could you give an interpretation of this finding in the discussion?

Response: Also, in general population (15) and in IBD (16) the PNPLA3 variant has been associated with elevations of liver enzymes. IBD patients have a greater risk of hepatic steatosis (OR 2.9 95%CI 1.1–7.8), with increased circulating alanine transaminase (16). Elevation of ALT values were related with increasing dose of PNPLA3 variant. 

In general population, the mechanism for elevation of liver enzymes associated with PNPLA3 variant is unclear (15). An addition concerning this has been included into discussion.

Reviewer #2: Dear authors, thanks for giving me the opportunity to review your manuscript.

In the manuscript, the authors aim at demonstrating how the presence of different alleles of PNPLA3 might influence clinical outcome and bile composition in patients with primary sclerosing cholangitis.

Despite the conflicting results of the other studies available in the literature and the numerous sample studied (not always available in case of rare diseases), in the reviewer's opinion, the rationale of the study is not convincing and, unsurprisingly the results confirm it. PNPLA3 has been implied in fibrogenesis and carcinogenesis and has been linked to hepatocarcinoma which is not a frequent tumor in PSC patients. Furthermore, PSC patients usually develop liver fibrosis as a consequence of cholestasis and in many cases, the fibrosis is confined to the bile ducts. The mechanisms by which PNPLA3 induces liver damage are not completely understood and, in the reviewer's opinion, the rationale for studying their alleles in PSC is weak. However, the evaluation of bile acid composition may be interesting in these patients, but a lot of confounding factors might be involved in this analysis (i.e. microbiota composition and IBD activity) which are not taken into account.

Response: The rationale to study PNPLA3 variant dose in PSC is based on the previously demonstrated role of PNPLA3 I148M variant exhibiting increased LPAAT activity and leads to increased cellular lipid accumulation and promote the disease progression to chronic liver disease under a large variety of harmful stimuli for the liver such as cholestasis or bacterial endotoxemia as suggested in PSC. Primary hepatic stellate cells from patients with the PNPLA3-rs738409 GG variant displayed significantly higher expression and release of pro-inflammatory cytokine (Bruschi FV, et al. The PNPLA3 I148M variant modulates the fibrogenic phenotype of human hepatic stellate cells. 

Hepatology. 2017;65:1875-90). 

In addition, in AIH patients, PNPLA3 I148M variant has shown to be associated with progressive disease despite the presence of steatosis being similar across all PNPLA3-rs738409 genotypes (Mederacke Y-S, et al. The PNPLA3 rs738409 GG genotype is associated with poorer prognosis in 239 patients with autoimmune hepatitis. Aliment Pharmacol Ther. 2020;51:1160-8).

In present study, we also wanted to analyze the possible impact of the I148M variant on biliary cholesterol, phospholipid, and BA composition, not previously reported.

The aim of the present study was to evaluate PNPLA3 allele dose effect on the clinical manifestations, disease progression and survival in a large PSC population, based on previous studies suggesting that the common I148M variant of the PNPLA3 gene is a risk factor for reduced survival (17). The authors suggested that genetic testing for the common PNPLA3 variant might improve diagnostic and therapeutic approaches in primary sclerosing cholangitis.

Furthermore, there are many punctuation errors in the manuscript and imprecisions.

Response: An English language edition has been done by Wiley Editing Services. A document of the manuscript edition is attached.

Introduction:

The introduction is not fluent and the rationale is not well explained.

I suggest rephrasing sentences in lines 76 and 81.

In line 88 I would add that the UDCA effect on LT-free survival and liver-related death has never been demonstrated.

Response: The rephrasing of the suggested sentences has been done.

A sentence: ‘However, the UDCA has not been demonstrated to have an effect on liver transplantation-free survival or on liver-related death’ have been added into introduction.

Methods:

The part on statistical analysis is too long and full of detail with respect to the data shown in the results section.

You mention a "compound" outcome in the results but I suppose composite outcomes have never been mentioned in the methods section. I suggest specifying the endpoints in this section.

Response: Statistics chapter has been written by biostatistician (HK). It has been shortened and describes now in detail what analysis have been done, and which tests have been used. The composite endpoint definition (CCA, HCC, liver related deaths, liver transplantation) has added in the Materials and Methods section.

Results:

line 162, specify the unit of measure of median values in brackets.

Response: The unit (years) has been added.

line 175, time to the first dilation since when? diagnosis? ERC?

Response: The time to first dilatation is defined as the time interval from first diagnostic ERC of the patient to the ERC when dilatation was first done. 

In table 2 I would have distinguished HCC from CCA, which is, in the reviewer's opinion, quite important to the aim of the manuscript.

Response: We totally agree and the table 2 has been corrected according to referee’s comment. HCC (n=11) is added as a separate end point, demonstrating no differences between groups, see also supplementary table 2.

An English editing is suggested.

Response: An English language edition has been done by Wiley Editing Services. A document of the manuscript edition is attached.

See the attached file Response to reviewers’ comments with supplementary tables.

---

## [Decision Letter · Decision Letter 1]

4 Oct 2022

PONE-D-21-38918R1PNPLA3 allele frequency has no impact on biliary bile acid composition and disease course in patients with primary sclerosing cholangitisPLOS ONE

Dear Dr. Färkkilä,

Thank you for submitting your manuscript to PLOS ONE. After careful consideration, we feel that it has merit but does not fully meet PLOS ONE’s publication criteria as it currently stands. Therefore, we invite you to submit a revised version of the manuscript that addresses the points raised during the review process.

We look forward to receiving your revised manuscript.

Kind regards,

Antonio De Vincentis

Academic Editor

PLOS ONE

Journal Requirements:

Reviewers' comments:

Reviewer's Responses to Questions

**Comments to the Author**

1. If the authors have adequately addressed your comments raised in a previous round of review and you feel that this manuscript is now acceptable for publication, you may indicate that here to bypass the “Comments to the Author” section, enter your conflict of interest statement in the “Confidential to Editor” section, and submit your "Accept" recommendation.

Reviewer #1: All comments have been addressed

Reviewer #2: All comments have been addressed

2. Is the manuscript technically sound, and do the data support the conclusions?

Reviewer #1: Yes

Reviewer #2: Yes

3. Has the statistical analysis been performed appropriately and rigorously? 

Reviewer #1: Yes

Reviewer #2: Yes

4. Have the authors made all data underlying the findings in their manuscript fully available?

Reviewer #1: Yes

Reviewer #2: Yes

5. Is the manuscript presented in an intelligible fashion and written in standard English?

Reviewer #1: Yes

Reviewer #2: Yes

6. Review Comments to the Author

Reviewer #1: Thank to the authors who addressed properly to all the raised questions. No further comments from this reviewer

Reviewer #2: In the abstract:

I would change “clinical manifestations” with clinical outcomes.

In the conclusions “Conclusions: PNPLA3 I148M variants have no a significant impact on either on bile composition including UDCA content, clinical manifestations, progression of liver fibrosis, the risk for hepatobiliary cancers, transplantation, or overall survival”. Check the grammar. For example on both before and after either and a before significant.

In the introduction:

- The reviewer would shorten the very long introduction. For example the sentence “However, PSC is associated with markedly increased risk of cholangiocarcinoma (CCA) (2), with lifetime risk 10%-20% (3,4), or 398 - 1000-fold compared to the general population (4,5). Several risk factors for CCA in patients with PSC, such as inflammatory bowel disease (IBD), especially ulcerative colitis (6), and older age at PSC diagnosis have been identified (4,7).” Is not so important in the context of the manuscript, in fact in the following part of the introduction you mention in patients with PNPLA3 variants have an increased risk of HCC and you don’t mention anymore CCA.

Some of the points explained in the introduction should be moved to the discussion which is shorter with respect to the introduction.

Results:

- Wording error in line 192.

7. PLOS authors have the option to publish the peer review history of their article (what does this mean?). If published, this will include your full peer review and any attached files.

Reviewer #1: No

Reviewer #2: No

---

## [Author Response · Author response to Decision Letter 1]

11 Oct 2022

Response to reviewers’ comments

Reviewer #1: Thank to the authors who addressed properly to all the raised questions. No further comments from this reviewer

Reviewer #2: In the abstract: I would change “clinical manifestations” with clinical outcomes.

The change has been done and the term ‘clinical manifestations’ has been replaced by ‘clinical outcomes’.

In the conclusions “Conclusions: PNPLA3 I148M variants have no a significant impact on either on bile composition including UDCA content, clinical manifestations, progression of liver fibrosis, the risk for hepatobiliary cancers, transplantation, or overall survival”. Check the grammar. For example on both before and after either and a before significant.

The conclusion in abstract has been rephrased as follow:” Conclusions: The PNPLA3 I148M variant had no significant impact on either on bile composition, including UDCA content, clinical outcomes, progression of liver fibrosis, hepatobiliary cancer risk, liver transplantation, or overall survival.

In the introduction:

- The reviewer would shorten the very long introduction. For example, the sentence “However, PSC is associated with markedly increased risk of cholangiocarcinoma (CCA) (2), with lifetime risk 10%-20% (3,4), or 398 - 1000-fold compared to the general population (4,5). Several risk factors for CCA in patients with PSC, such as inflammatory bowel disease (IBD), especially ulcerative colitis (6), and older age at PSC diagnosis have been identified (4,7).” Is not so important in the context of the manuscript, in fact in the following part of the introduction you mention in patients with PNPLA3 variants have an increased risk of HCC and you don’t mention anymore CCA.

The introduction has been shortened as suggested by reviewer #2: the sentence in rows 50-53 has been deleted. See also below the response to next reviewer’s comment.

The patients with PNPLA3 variants have been reported to have an increased risk of HCC and but there is no published data that the variants would increase the risk for CCA.

Some of the points explained in the introduction should be moved to the discussion which is shorter with respect to the introduction.

The following sentence from introduction (rows 81-87) has been moved into the discussion:” A recent study evaluating plasma BA profiles in PSC to predict hepatic decompensation, demonstrated that the risk was associated with increased concentration of total BA and conjugated fraction of many BA. In contrast, higher glycine: taurine conjugation ratios were protective (24). Intestinal dysbiosis has been described in PSC patients with and without IBD (25). The effect of the PNPLA3 rs738409 variant on gut microbiota has been evaluated only in a few studies. Monga Kravetz A et al. (26) demonstrated that this PNPLA3 variant caused an increased on Firmicutes/Bacteroides-ratio in obese youth with NAFLD.”

Results:

- Wording error in line 192.

The spelling mistake has been corrected.

---

## [Editor Report · Decision Letter 2]

20 Oct 2022

PNPLA3 allele frequency has no impact on biliary bile acid composition and disease course in patients with primary sclerosing cholangitis

PONE-D-21-38918R2

Dear Dr. Färkkilä,

We’re pleased to inform you that your manuscript has been judged scientifically suitable for publication and will be formally accepted for publication once it meets all outstanding technical requirements.

Kind regards,

Antonio De Vincentis

Academic Editor

PLOS ONE

---

## [Editor Report · Acceptance letter]

22 Nov 2022

PONE-D-21-38918R2 

 PNPLA3 ALLELE FREQUENCY HAS NO IMPACT ON BILIARY BILE ACID COMPOSITION OR DISEASE COURSE IN PATIENTS WITH PRIMARY SCLEROSING CHOLANGITIS 

Dear Dr. Färkkilä:

I'm pleased to inform you that your manuscript has been deemed suitable for publication in PLOS ONE. Congratulations! Your manuscript is now with our production department. 

Kind regards, 

on behalf of

Dr. Antonio De Vincentis 

Academic Editor

PLOS ONE